# The Impact of Kidney Biopsy for Fabry Nephropathy Evaluation on Patients’ Management and Long-Term Outcomes: Experience of a Single Center

**DOI:** 10.3390/biomedicines10071520

**Published:** 2022-06-27

**Authors:** Elena-Emanuela Rusu, Diana-Silvia Zilisteanu, Lucia-Mihaela Ciobotaru, Mihaela Gherghiceanu, Alexandru Procop, Ruxandra-Oana Jurcut, Adriana Octaviana Dulamea, Bogdan Marian Sorohan

**Affiliations:** 1Department of Nephrology, “Carol Davila” University of Medicine and Pharmacy, 020021 Bucharest, Romania; diana.zilisteanu@umfcd.ro (D.-S.Z.); lucia.ciobotaru@gmail.com (L.-M.C.); mihaela.gherghiceanu@umfcd.ro (M.G.); ruxandra.jurcut@umfcd.ro (R.-O.J.); adriana.dulamea@umfcd.ro (A.O.D.); bogdan.sorohan@yahoo.com (B.M.S.); 2Department of Nephrology, Fundeni Clinical Institute, 022328 Bucharest, Romania; 3“Victor Babes” National Institute for Research and Development in Pathology and Biomedical Sciences, 050097 Bucharest, Romania; 4Anatomic Pathology, Fundeni Clinical Institute, 022328 Bucharest, Romania; procop_alex@yahoo.com; 5Department of Cardiology, Emergency Institute for Cardiovascular Diseases “Prof. Dr. C. C. Iliescu”, 022328 Bucharest, Romania; 6Department of Neurology, Fundeni Clinical Institute, 022328 Bucharest, Romania; 7Department of Uronephrology and Kidney Transplantation, Fundeni Clinical Institute, 022328 Bucharest, Romania

**Keywords:** Fabry disease, kidney biopsy, Fabry nephropathy, outcome, early indicator

## Abstract

Background: Fabry disease (FD) is a rare lysosomal storage disease causing progressive loss of target organ function. All renal cell types are involved from the early stages, even before clinical signs can be detected. FD-specific therapies can stop/mitigate disease progression. Thus, it is important to validate early markers of renal lesions so that they can be adopted as criteria for timely treatment initiation. Materials and methods: We retrospectively analyzed and extensively evaluated 21 FD case patients; this evaluation included a kidney biopsy. We looked for the influence of pathological findings on the management of FD patients. In addition, we investigated the association between general and FD-specific features and long-term patients’ outcomes. We defined a combined endpoint as being at least one of the following: 50% decrease of estimated glomerular filtration rate (eGFR) from baseline, kidney failure (KF), end-stage kidney disease (ESKD), or death and mortality. Results: Our cohort of 21 FD patients (11 males and 10 females) was stratified according to the presence of the combined endpoint: group 1 (*n* = 15) included patients without the combined endpoint, while group 2 (*n* = 6) patients reached the combined endpoint outcome. Patients from group 2 presented lower mean baseline eGFR (72.2 ± 38.7 mL/min/1.73 m^2^ vs. 82.5 ± 26.4 mL/min/1.73 m^2^) without statistical significance (*p* = 0.44), but significantly (*p* = 0.22) higher median baseline proteinuria (2.7 g/24 h vs. 0.4 g/24 h). Specific lysosomal deposits were identified in all patients. Segmental sclerosis was present in all patients with the combined endpoint and in only 33% of patients without the combined endpoint (*p* = 0.009). Global sclerosis and interstitial fibrosis were present in both groups, with no significant differences. A total of 15 out of the 16 treatment-naïve patients (7 males and 9 females) started FD-specific therapy after kidney biopsy. Treatment was initiated in all male FD patients and in 8 female patients. In 2 females, pathological findings in kidney biopsy offered important reasons to start FD treatment, although specific criteria of the Romanian protocol for prescription of FD-specific therapy were still not fulfilled. Cox univariate analysis showed that every increase in 24 h proteinuria with 1 g is associated with a 65% risk of developing the combined endpoint (HR = 1.65; 95%CI: 1.05–2.58; *p* = 0.02), and that the presence of segmental sclerosis increased the risk of developing the combined endpoint by 51.3 times (HR = 51.3; 95% CI: 95% CI: 1.67–103.5; *p* = 0.01). Kaplan–Meier analysis showed that the cumulative risk of developing the combined endpoint was higher in patients in whom segmental sclerosis (100% vs. 0%, log-rank test, *p* = 0.03) was present. Conclusions: Histological evaluation is an important tool for the detection of early kidney involvement and provides additional support to the early initiation of FD-specific therapy. Presence of segmental sclerosis can predict the long-term outcomes of kidney disease deterioration and mortality and may be used as an early indicator of disease progression. Additionally, in the absence of other criteria according to current guidelines, specific FD renal lesions as revealed by kidney biopsy might become a distinct criterion to initiate FD therapy.

## 1. Introduction

Fabry disease (FD) is a rare X-linked recessive lysosomal storage disease caused by mutations in the α-galactosidase A (GLA) gene, leading to a deficient activity of α-GLA [1]. The lysosomal accumulation of glycosphingolipids—particularly globotriaosylceramide (GL3)—within lysosomes in a variety of cell types—including capillary endothelial cells, renal, cardiac and nerve cells—results in several clinical signs and symptoms and substantial morbidity and mortality [1,2]. Classically affected hemizygous males with no residual α-GLA activity may present a wide spectrum of clinical manifestations, such as neuropathic pain, cornea verticillata, angiokeratoma, hypertrophic cardiomyopathy, cardiac rhythm disturbances, proteinuria, progressive renal failure, transient ischemic attack and stroke, as well as cochleo-vestibular signs [2]. Heterozygous females may have heterogeneous symptoms due to X-chromosome random inactivation (lyonization); these symptoms range from very mild to severe [3,4]. In the kidney, it is GL3 that predominately accumulate in podocytes and are responsible for the early development of proteinuria. Nevertheless, all renal cell types within the kidney can be affected, even in patients with normal glomerular filtration rate (GFR) and minimal proteinuria [5,6,7]. Proteinuria, which has long been considered the first clinical manifestation of Fabry nephropathy, may start as early as at 10 years of age [8]. All these FD renal lesions generate progressive kidney failure and significantly limit life expectancy in affected patients [1,9]. Renal lesions are found in both hemizygous (male) and heterozygous (female) patients. Renal symptoms in females are typically milder and delayed by 2 to 3 decades, but there is considerable variability [6]. A significant proportion of female patients suffer from significant complications, including 40% with clinical renal disease (mainly proteinuria) and about 15% with serious renal events [4]. Thus, Fabry disease not only has a highly variable symptomatology, but also a different rate of progression [10]. The gold standard for diagnosis of FD nephropathy was defined as characteristic storage on electron microscopy in a kidney biopsy specimen in the absence of medication that may induce similar storage [11]. Kidney biopsy is useful in all patients with any degree of proteinuria/albuminuria and/or renal dysfunction to assess the presence of glomerulosclerosis and interstitial damage because of their prognostic significance [12,13]. In addition, in individuals with a genetic variant of unknown significance without classical FD manifestations, or when other diseases could be responsible for the renal manifestations, a kidney biopsy with electron microscopy examination should be performed to confirm or exclude the diagnosis of FD nephropathy [11,14]. Patients with FD diagnosis should start specific treatment as soon as possible in order to avoid disease progression and irreversible organ damage [15,16]. FD-specific therapy includes enzyme replacement therapy (ERT) with agalsidase beta and agalsidase alfa, and pharmacological chaperone migalastat for patients with amenable mutations [16,17].

This retrospective, single-center, cohort study evaluated firstly the impact of Fabry nephropathy diagnosed by kidney biopsy on the management of patients, and secondly the association between general and FD-specific features and long-term patients’ outcomes, defined as 50% decrease of eGFR from baseline, reaching end-stage kidney disease (ESKD), kidney failure (KF) and mortality. A total of 21 patients with Fabry disease who underwent kidney biopsy were included in this study. 

We hereby present a retrospective, single-center, cohort study which included a total of 21 patients with Fabry disease who underwent kidney biopsy. The aim of this analysis is to evaluate the impact of kidney biopsy findings on the long-term outcome and on the therapeutic management of FD patients.

## 2. Materials and Methods

### 2.1. Study Participants

From a total of 48 patients with genetically confirmed Fabry disease, our retrospective study included 21 patients (11 males and 10 females) who underwent kidney biopsy. The patients were evaluated and monitored in the Expert Center for Rare Diseases of Reno-Urinary System, Fundeni Clinical Institute, Bucharest, during 2015–2021. All 21 patients had comprehensive clinical, biological and imaging assessment for manifestations of Fabry disease as part of the follow-up protocol of our center. The study conformed to current clinical practice and anonymized patient data were analyzed. The study was conducted in accordance with the principles of the Helsinki Declaration and was approved by the Institutional Ethics Committee of the Fundeni Clinical Institute (No. 28641/2022). 

### 2.2. Clinical Assessments and Study Endpoints

Patients were evaluated by the Fabry multidisciplinary team of the Fundeni Clinical Institute, Bucharest and the Emergency Institute for Cardiovascular Diseases, “Prof. Dr. C. C. Iliescu”, Bucharest in order to reveal the involvement of target organs in Fabry disease, to establish the indication for FD-specific treatment initiation and to monitor disease progression. We performed family genetic screening to identify other affected family members. 

Demographic, clinical and biological data at baseline and during follow-up; comorbidities; past and current treatment; severe renal clinical events; and deaths were recorded.

All determinations of the level of α-GLA enzyme activity and lyso Gb3, and GLA gene analyses were performed at ARCHIMED life Science GmbH (Vienna, Austria). GLA mutations were classified according to recommendations of the American College of Medical Genetics and Genomics (ACMG): Pathogenic variants: variants that are causative of a disease;Likely pathogenic variants: variants where the data support a high likelihood that it is pathogenic;Variants of uncertain significance (VUS): a genetic variant with unknown or questionable impact on clinical phenotype;Likely benign: variant where the data support a high likelihood that it is benign;Benign: a variant that is not considered to be the cause of the disease [18].

The evaluation of Fabry nephropathy included measurement of serum creatinine and the estimated glomerular filtration rate (eGFR), determination of the urine albumin: creatinine ratio (UACR) in spontaneous urine and measurement of protein in 24-h urine collection, and kidney biopsy. The eGFR was calculated using Chronic Kidney Disease Epidemiology Collaboration (CKD-EPI) equation for adults and Schwartz formula for children [19,20]. Chronic kidney disease (CKD) stages were classified as follows: stage 1 for GFR ≥ 90 mL/min/1.73 m^2^; stage 2 for GFR 60–89 mL/min/1.73 m^2^; stage 3 GFR 30–59 mL/min/1.73 m^2^; stage 4 for GFR 15–29 mL/min/1.73 m^2^; stage 5 for GFR < 15 mL/min/1.73 m^2^ or treated by dialysis [21]. A urinary albumin: creatinine ratio (UACR) < 30 mg albumin/g creatinine was considered normal to mildly increased, UACR of 30–300 mg/g and UACR > 300 mg/g were defined as microalbuminuria and macroalbuminuria, respectively [21]. The imaging workup included ultrasound for detection of cysts, vascular lesions and chronic parenchymatous changes. In patients with elevated blood pressure, an ambulatory 24-h blood pressure measurement was performed. Fabry disease severity was also assessed using the Mainz Severity Score Index (MSSI) [22].

Kidney biopsy specimens were studied by light microscopy and electron microscopy. The pathological findings referred to glomerular sclerotic lesions (global sclerosis or segmental sclerosis), interstitial fibrosis, tubular atrophy, vessel changes (hyaline and arteriopathy), detection of GL3 deposits in the podocytes, tubular cells and glomerular endothelial cells. Pathological findings were scored as present or absent. Global sclerosis was scored as present when ≥50% of glomeruli were sclerotic, and interstitial fibrosis was scored as present when ≥25% of kidney interstitium was affected. 

Additionally, existence of neurological involvement was assessed by clinical exam, electroneurographic exam and brain magnetic resonance, whereas heart manifestations were evaluated by echocardiography, electrocardiogram, ECG Holter and cardiac magnetic resonance. 

Baseline was defined as the moment when kidney biopsy was performed. The follow-up period was variable for every patient from the baseline to the last visit into the clinic, with a mean of 47.7 ± 19.1 months and a mandatory condition for monitoring for at least 12 months.

All the patients included in the present analysis were monitored for at least 12 months.

The study endpoints were defined as: mean change of eGFR from baseline, mean change of 24 h proteinuria from baseline, a decrease of eGFR with 50% from baseline, ESKD kidney failure, or death and mortality. Additionally, due to the small number of patients and the low frequency of events, we developed a combined endpoint which included at least one of the following: a decrease of eGFR with 50% from baseline, ESKD KF, or death and mortality.

### 2.3. Statistical Analysis

Data were presented as frequencies with percentages for categorical variables, mean with standard deviation for continuous parametric data and median with interquartile range for those continuous nonparametric. For variable comparison, Chi-square or Fisher exact tests were used as appropriate for categorical data, Student’s *t*-test for continuous parametric data and Mann–Whitney U for continuous nonparametric data. Paired-samples *t*-test was used to evaluate the mean difference of a variable between baseline and last time of follow-up, where baseline was defined as the time of kidney biopsy. Cox regression was used to test the HR which was significant at comparison analysis. Kaplan–Meier curves were used to determine the cumulative risk of the combined endpoint and log-rank test was applied to analyze the cumulative risk difference. Violin plot graphs with individual values were utilized to highlight mean differences. A *p* value < 0.05 was considered statistically significant. 

Data statistical analysis and figures were performed using SPSS version 26 (SPSS Inc., Chicago, IL, USA), STATA version 14 (StataCorp, College Station, TX, USA) and GraphPad Prism version 9.3.1 (1992–2021 GraphPad Software, LLC, San Diego, CA, USA). 

## 3. Results

### 3.1. Patients

#### 3.1.1. General Clinical and Biological Parameters

The baseline characteristics of the study group are presented in Table 1. All 21 patients (11 males and 10 females) had pathogenic mutations for Fabry disease. A pathogenic variant, classic or late-onset, was identified in all 21 patients (11 males and 10 females):Classic: c.797A > C in 5 patients, c.485G > A in 3 patients, c.779G > A, c.295C > T, c.836A > G each in the case of 2 patients, and c.863C > A, c.671delA, c.1224del66, c.1228A > G, c.141G > A, c.1121_1123delAAG each in the case of 1 patient;Late-onset: c.644A > G (N215S) in 1 patient.

Ten patients were index cases, while the others were diagnosed by family screening.

The mean period of follow-up was 47.7 ± 19.1 months. The mean age at the time of renal biopsy was 43.7 ± 14.2 years, with significantly younger males than females (33.5 ± 12.9 versus 47.6 ± 11.7 years, *p* < 0.05). Eight patients (38.1%) had arterial hypertension at baseline and one other patient developed it during follow-up. Two patients had diabetes mellitus (DM), a 10-year-old boy with type 1 DM and a 61-year-old female with type 2 DM. Fifteen patients (71.4%—8 males and 7 females) had CKD stage 1 or 2, five patients had CKD stage 3 (28.5%—2 males and 3 females) and one male patient had CKD stage 4.

Our cohort was divided into two groups according to the presence or absence of the combined endpoint, defined as at least one of the following: decrease of eGFR with 50% from baseline, or ESKD KF, or mortality. Group 1 (*n* = 15) included patients without the combined endpoint and group 2 (*n* = 6) patients who satisfied the combined endpoint outcome. Table 2 provides a comprehensive overview of the general and specific Fabry features, kidney biopsy findings and treatment, and presents study outcomes according to the combined endpoint. 

In group 1, there were more females (60%) than males, while in group 2 there were more males (83.3%). Overall baseline mean eGFR was 79.6 ± 26.7 mL/min/1.73 m^2^, mean baseline eGFR in group 1 was 82.5 ± 26.4 mL/min/1.73 m^2^, and mean baseline eGFR in group 2 was 72.2 ± 38.7 mL/min/1.73 m^2^. Overall eGFR at last follow-up was 68.0 ± 37.3 mL/min/1.73 m^2^, with a significant lower (*p* = 0.003) mean eGFR at last follow-up in group 2 (32.5 ± 38.54 mL/min/1.73 m^2^) compared to group 1 (82.2 ± 26.4 mL/min/1.73 m^2^). Eight patients presented normal UACR, seven patients showed microalbuminuria, and six patients presented macroalbuminuria. Overt proteinuria (> 0.3 g/24 h) was noticed in 8 patients (5 males and 3 females), and 7 out of 8 patients presented proteinuria > 1 g/24 h. Median proteinuria was significantly higher in group 2 (2.7 g/24 h, range 0.2–3.7) than in group 1 (0.4 g/24 h, range 0.1–0.4). The mean overall follow-up period was 47.7 ± 19.1 months, with similar values between the two groups (49.2 ± 19.7 months in group 1 and 44.0 ± 18.4 months in group 2).

#### 3.1.2. Fabry Features

Table 3 shows the genotype, baseline level of α-GLA activity and plasma globotriaosylsphingosine (lyso-GL3), as well as MSSI score of our study cohort.

A pathogenic variant, classic or late-onset, was identified in all 21 patients (11 males and 10 females):Classic: c.797A > C in 5 patients, c.485G > A in 3 patients, c.779G > A, c.295C > T, c.836A > G each in the case of 2 patients, and c.863C > A, c.671delA, c.1224del66, c.1228A > G, c.141G > A, c.1121_1123delAAG each in the case of 1 patient;Late-onset: c.644A > G (N215S) in 1 patient.

Ten patients were index cases, while the others were diagnosed by family screening.

Alfa-GLA activity was reduced in 20 patients, median 0.4 µmol/L/h (range 0.1–1.2), with lower values in group 2 (0.3 µmol/L/h, range 0–0.4 µmol/L/h) compared to group 1 (0.6 µmol/L/h, range 0.2–1.5 µmol/L/h), but without statistical significance, *p* = 0.13. The median plasma globotriaosylsphingosine (lyso-GL3) levels were 6.8 ng/mL (4.9–22.5 ng/mL) in group 1 and 27 ng/mL (9.2–105.5 ng/mL) in group 2 (*p* = 0.22) (Table 2).

The MSSI scores of patients in the combined endpoint group were similar to those in the stable renal function group. Other major organ involvement damage included neurological involvement (in 90.5% of patients) and hypertrophic cardiomyopathy (in 42.8% of patients), without significant differences between the two groups. Alfa-GLA activity was reduced in 20 patients, median 0.4 µmol/L/h (range 0.1–1.2), with lower values in group 2 (0.3 µmol/L/h, range 0–0.4 µmol/L/h) compared to group 1 (0.6 µmol/L/h, range 0.2–1.5 µmol/L/h), but without statistical significance, *p* = 0.13. The median plasma globotriaosylsphingosine (lyso-GL3) levels were 6.8 ng/mL (4.9–22.5 ng/mL) in group 1 and 27 ng/mL (9.2–105.5 ng/mL) in group 2 (*p* = 0.22). 

### 3.2. Kidney Biopsy Findings

Histological findings are presented in Table 2 and Table 4. The median glomeruli number evaluated by light microscopy was 8 (5–10). Segmental sclerosis was observed in 11 (52.4%) patients, 7 out of 11 males (4 males with CKD stage 1, 2 males with CKD stage 2, and 1 male with CKD stage 4), and 4 out of 10 females (2 females with CKD stage 2, and 2 females with CKD stage 3). Segmental sclerosis was found even in two male patients with stage 1 CKD without albuminuria/proteinuria (patient No. 7 and patient No. 19). Segmental sclerosis was present in all patients with the combined endpoint and in only 33% of patients without the combined endpoint (*p* = 0.009). Global sclerosis was present in 6 (28.6%) patients: 3 females (1 with CKD stage 2 and 2 with CKD stage 3) and 3 males (1 with CKD stage 1, 1 with CKD stage 2, and 1 with CKD stage 3), with no significant difference between the two groups. Interstitial fibrosis was similar in patients with the combined endpoint (66.7%) compared to patients without the combined endpoint (40%), *p* = 0.36. Other light microscopy changes were observed as follows: glomerular hyaline in 4 patients, tubular atrophy in 8 patients, and arteriopathy in 8 patients. Podocyte GL3 deposits and tubular epithelium cell GL3 deposits were present in all patients, and glomerular endothelial cell GL3 deposits were revealed in 20 out of 21 patients (Figure 1). 

### 3.3. Treatment

The Romanian national therapeutic protocol for prescription of FD-specific therapy states that the eligible patients for the initiating of the therapy are: males over 16 years old with confirmed FD, boys over 10–13 years old with significant manifestations or asymptomatic, and females of all ages with significant manifestations or with documented progressive organ involvement. Significant renal manifestations (renal criteria) are represented by persistent proteinuria over 300 mg/24 h and/or eGFR < 80 mL/min/1.73 m^2^ [23]. 

Within the National Treatment Program for Rare Diseases, the following drugs are available for FD-specific therapy: agalsidase beta (the first drug used in Romania, used since 2006), migalastat (available since 2019) and agalsidase alfa (available since 2021).

In our cohort, 19 patients were treated with agalsidase beta. One of them was switched from agalsidase beta to migalastat. One naïve patient was treated with migalastat.

At the moment of kidney biopsy, five patients were treated with enzyme replacement therapy (ERT), beta agalsidase beta 1 mg/kg every 2 weeks, for 75 ± 57 months (range 12–144). Three of these patients were in group 1, and two patients were in group 2. Sixteen patients were treatment-naïve at the moment of kidney biopsy. Fifteen patients started Fabry disease-specific therapy after kidney biopsy. A comparison analysis of patients who started the treatment before kidney biopsy and those who began the treatment after kidney biopsy showed no differences in terms of the combined EP and histological lesions (Table 5).

Considering Romanian national protocol for initiating FD-specific therapy, all male patients fulfilled the criteria. Our cohort included two pediatric patients. One 17-year-old boy presented severely clinical and histological kidney involvement. The other is a 10-year-old boy presented with a family history of severe kidney involvement in males, undetectable αGLA activity, plasma lysoGb3 101.1 ng/mL, acroparesthesia and type 1 diabetes as significant co-morbidity, which could contribute to renal damage. In the case of this second boy, kidney biopsy was mandatory and showed specific FD lesions histological damage at kidney biopsy (podocyte, tubular and glomerular endothelial cell deposits), while and excluded lesions specific to diabetes. Due to the fact that the 10-year-old boy also had type 1 diabetes, the kidney biopsy was needed to assess whether there were superimposed lesions specific to diabetes. Both patients received indication for treatment and started ERT according to the national protocol and in accordance with the recommendations for treatment of FD in pediatric patients [10,24]. 

Regarding therapy naïve females (*n* = 9), 4 patients (40%) fulfilled the renal criteria, and 5 patients did not fulfill renal criteria according to the Romanian national protocol for initiating FD-specific therapy. Of these 5 patients, 2 patients presented criteria for other organs involvement, while 3 patients (mean age 37.7 years) did not fulfill any clinical, biological and imaging criteria for any other organ in order to initiate FD-specific therapy. We emphasize that, even in our 5 female patients without renal criteria for FD-specific therapy, kidney biopsy showed Fabry specific lesions: podocyte, tubular and glomerular endothelial cell deposits in all cases. These FD-specific changes were associated with interstitial fibrosis and arteriopathy in one case, and with tubular atrophy in another case. Our 3 female patients without any organ criteria for FD-specific therapy were monitored every 6 months. In two patients we noticed a decline of eGFR during follow up, without falling below the threshold of 80 mL/min/1.73 m^2^. In these 2 cases, based on the decreasing trend of renal functions, on the histological findings and on family history (other relatives with FD and significant renal involvement), initiation of FD-specific therapy was approved, although the specific criteria of the Romanian protocol were still not fulfilled. Thus, in two of nine (22%) treatment-naïve females from our cohort, kidney biopsy was crucial for FD-specific therapy initiation. One of our female patients is still untreated and is followed up regularly. 

Fourteen patients started ERT (beta-agalsidase beta, at a dose of 1 mg/kg every 2 weeks) after 8.15 ± 7.18 months from baseline, and one patient received migalastat 5 months later. 

We did not register any renal or other events before the start of FD-specific therapy. During follow-up, one patient was switched from beta-agalsidase beta to migalastat after 26 months. 

Angiotensin-converting enzyme inhibitors or angiotensin receptor blockers were given to 9 patients (42.8%) at baseline. The indications for RAAS inhibitor treatment were the arterial hypertension (presented in 8 patients) and increased proteinuria. During follow-up, another patient developed arterial hypertension and was treated with angiotensin receptor blocker. 

### 3.4. Outcomes

The outcomes of the study cohort were presented in Table 2. At the end of the follow-up, an eGFR decline by 50% from baseline was observed in five patients (23.8%), ESKD KF in three patients (14.3%), and death was recorded for two patients (9.5%). Six patients (28.6%) had the combined endpoint. 

ESKD KF appeared in two males (patients No. 1 and No. 5) and one female (patient No. 15), at the age of 18, 47 and 41 years, respectively. These patients had advanced Fabry nephropathy at baseline, as follows: two had CKD stage 3 (one male and one female) and one male had CKD stage 4. Additionally, all these cases had severe proteinuria at baseline (range between 3.26–4.5 g/24 h). Initially, all ESKD KF patients were included in hemodialysis, subsequently Of these, the two KF males subsequently had a living donor kidney transplant, with an excellent evolution of renal graft after about 5 years, respectively 6 months after kidney transplant. All KF patients continued FD-specific treatment in order to prevent further organ damage. In terms of mortality, two male patients died at 39 and 58 years old, respectively, and the cause of death was stroke in both cases. 

The main outcome of our study cohort was the combined endpoint. The features of patients with the combined endpoint were described in Table 2. We found that patients who developed the combined endpoint were more frequently males (*p* = 0.06) (Figure 2A), had significantly higher levels of 24 h proteinuria at baseline (*p* = 0.02) (Figure 2B), had a significantly higher percentage of segmental sclerosis on kidney biopsy (*p* = 0.009) (Figure 2C) and a significantly lower mean eGFR at the end of the follow-up period (*p* = 0.003) (Figure 2D).

Mean change of eGFR and 24 h proteinuria from baseline during the period of follow-up (47.7 ± 19.1 months) were presented in Figure 3. We found a mean eGFR decline from baseline to the end of follow-up of −11.6 mL/min/1.73 m^2^ (95% CI: −0.5–+23.7, *p* = 0.06), which was at the limit of statistical significance (Figure 3A) and a non-significant change in the mean 24 h proteinuria (+0.1, 95%CI: −0.4–+0.4, *p*= 0.94) (Figure 3B). However, for 15 patients included in group 1, eGFR remained stable during the follow-up period (82.5 ± 21.3 mL/min/1.73 m^2^ at baseline versus 82.2 ± 26.4 mL/min/1.73 m^2^).

Cox univariate analysis was performed to test the association of 24 h proteinuria and segmental sclerosis with the combined endpoint. According to the analysis, we observed that every increase in 24 h proteinuria with 1 g is associated with a 65% risk of developing the combined endpoint (HR = 1.65; 95% CI: 1.05–2.58; *p* = 0.02), and that the presence of segmental sclerosis increased the risk of developing the combined endpoint by 51.3 times (HR = 51.3; 95% CI: 95% CI: 1.67–103.5; *p* = 0.01).

Following the Kaplan–Meier analysis (Figure 4A,B), we found that the cumulative risk of developing the combined endpoint was higher, but not statistically significant in patients with baseline proteinuria ≥ 1 g/24 h at 60 months and at the end of follow-up (59% vs. 29%, log-rank test, *p* = 0.15) (Figure 4A), and that the cumulative risk is significantly higher in patients with the presence of segmental sclerosis at the aforementioned periods of follow-up (100% vs. 0%, log-rank test, *p* = 0.03) (Figure 4B).

## 4. Discussion

The current study included 21 cases with Fabry disease, in which kidney biopsy was performed in order to accurately assess the renal involvement. Renal involvement was assessed through kidney biopsy. This is the largest clinical and histological case study of Romanian patients with FD. The evaluation of FD-specific lesions as well as nonspecific pathological findings offers valuable information about the severity of kidney damage, the prognosis and the evolution of the Fabry nephropathy. 

Most of our patients (15 out of 21: 8 males and 7 females) presented mild nephropathy (CKD stage 1 and 2). A total of 11 of them (73.3%) were normoalbuminuric (8 patients: 4 males and 4 females) and 3 patients presented microalbuminuria (1 male and 2 females). Pathologic findings of FD-specific lesions were reportedly identified in patients with mild nephropathy and normoalbuminuria or microalbuminuria in both therapeutically naïve and previously treated patients. Thus, in 8 of our patients without FD-specific treatment (3 males and 5 females with CKD stage 1 and 2), we noticed Fabry nephropathy with pathologic GL3 deposits in podocytes, tubular cells and glomerular endothelial cells. In addition, 3 ERT-treated patients (2 males with CKD stage 1 and 1 female with CKD stage 2) showed GL3 deposits, regardless of ERT. 

Previous studies showed that histological confirmation of kidney involvement precedes clinical signs in early stages of Fabry nephropathy [6,7,25,26,27]. In a recent study, Kim I.Y. et al. reported the clinical and pathologic findings from a group of 9 patients receiving ERT and another group of 6 patients before treatment initiation. In the before-treatment group, segmental foot process effacement and pathologic GL3 accumulation were observed in renal tissue in patients with normoalbuminuria [26]. 

Thus, an important observation of our study that is concordant with other studies that evaluated kidney involvement in FD through kidney biopsy is that specific Fabry disease findings were observed in both genders, even in patients with normoalbuminuria/microalbuminuria, demonstrating that kidney biopsy is a very important tool for early Fabry nephropathy assessment.

Moreover, nonspecific and chronic findings were present in kidney biopsy specimens of our 15 patients with mild nephropathy (with CKD stage 1 and 2) and normoalbuminuria/microalbuminuria. Thus, segmental sclerosis was observed in 3 out of 15 (20%) patients. Additionally, interstitial fibrosis was a histological finding in 3 of these 15 patients with mild nephropathy. Tubular atrophy was present in 3 cases, glomerular hyaline in 2 cases and arteriopathy in 1 case. Glomerular sclerosis is considered an early finding in Fabry nephropathy and also has a prognostic implication [25,27,28,29]. Fogo A.B. et al. observed that glomerular sclerotic lesions were present in 63% of patients with eGFR > 60 mL/min/1.73 m^2^ and that there were no significant histological differences between genders in early CKD. One of their conclusions was that clinical assessments of Fabry nephropathy in early stages lack sensitivity and may delay the initiation of ERT, emphasizing the need for kidney biopsy [25]. Tøndel C. et al. revealed renal lesions in biopsies from nine children (7 boys and 2 girls, age range 7 to 18 years old) with Fabry disease who had normal GFR. All the children presented GL3 deposits in podocytes and distal tubules and segmental foot process effacement, consistent with podocyte injury in all patients, including those with normoalbuminuria. Four of the nine children had arteriopathy and three of the nine children had focal segmental glomerulosclerosis. Through assessment by kidney biopsy, they concluded that glomerular and vascular lesions are present before progression to overt proteinuria and before the decline of glomerular filtration rate [7]. 

In our study, six patients (28.6%) reached the combined endpoint that included at least one of the following: the decrease of eGFR with 50% from baseline, ESKD KF or and mortality after the follow-up period. The risk of developing the combined endpoint was associated with segmental sclerosis at kidney biopsy and baseline 24 h proteinuria. The presence of segmental sclerosis in kidney biopsy specimen was a strong significant risk factor for developing the combined endpoint (Log Rank test, *p* = 0.03). This finding is consistent with other studies that showed that glomerular sclerosis predicts proteinuria, CKD stage and renal prognosis [25,27,29]. Germain D.P. et al. reported an accelerated eGFR decline in Fabry patients with segmental and global glomerular sclerosis in >50% of their glomeruli despite ERT treatment [30].

Other studies demonstrated that proteinuria is a significant risk factor for renal disease progression in FD patients treated with ERT [31,32]. D.G. Warnock et al. showed that urinary protein: creatinine ratio was the most powerful determinant of Fabry nephropathy progression in adults treated with agalsidase beta [26]. Additionally, in a retrospective study of long-term outcomes of ERT in FD, M. Arends et al. observed that the presence of proteinuria at baseline resulted in an additional decline in eGFR [32]. In our study, we found that every increase in baseline 24 h proteinuria with 1 g was associated with a 65% risk of developing combined endpoint (HR = 1.65; CI:1.05–2.58; *p* = 0.02).

Our results also have implications regarding the management of FD patients’. Thus, in our experience, histological assessment of Fabry nephropathy can influence the decision to start FD-specific therapy.

In the past years, due to increased experience with long-term outcome on ERT in patients with FD, the therapeutic approach has moved towards early ERT initiation. Thus, some retrospective analysis of outcome in FD patients treated with ERT showed that baseline eGFR predicts FD progression on ERT [32,33,34]. An increased efficacy of enzyme replacement therapy and the long-term benefits of early therapy were observed in patients who initiated ERT treatment at a younger age and with less kidney involvement [28,35,36,37,38]. 

Current international recommendations and the Romanian national protocol regarding treatment initiation of enzyme replacement therapy mainly focus on the need of treatment initiation in patients with established renal clinical signs [10,23,39]. However, increasing evidence about the benefits of early initiation of FD-specific therapy has led global experts to focus on using early indicators of disease progression in order to initiate FD-specific treatment. [15,17,40]. 

The question that is rising is how early to initiate the pharmacologic therapy [41]. In the international report PRoposing Early Disease Indicators for Clinical Tracking in Fabry Disease (PREDICT-FD) modified Delphi initiative expert consensus, the early indicators of clinical kidney damage include microalbuminuria, glomerular hyperfiltration and podocyte GL3 inclusions in the presence of other renal lesions, such as signs of glomerulosclerosis or vasculopathy, which may occur even in patients without microalbuminuria [15,40]. 

In our cohort, all males presented a classic pathogenic mutation and received Fabry disease specific-therapy. In these cases, the kidney biopsy provided us with important information regarding the prognosis of the disease, while in the case of those already treated it provided us with information regarding the response to treatment. 

The therapeutic approach in FD female patients was more difficult. According to the Romanian national protocol, kidney signs and symptoms that are currently used to guide the initiation of FD-specific therapy in females are markers of established kidney damage [23]. In the present study, 9 females had a kidney biopsy performed before FD-specific treatment initiation and in all cases histological examination revealed specific FD lesions. Five of these patients did not fulfill the Romanian national protocol renal criteria to initiate FD-specific treatment in females. Moreover, three of these patients did not present other organs involvement, and in two of them biopsy-proven specific lesions allowed us to recommend FD-specific treatment initiation. Therefore, evidence of specific FD lesions in kidney biopsy was a strong reason to support the recommendation to initiate FD-specific treatment. According to Romanian national protocol, kidney signs and symptoms that are currently used to guide the initiation of FD-specific therapy in females are markers of established kidney damage. We found no differences regarding the combined EP and histological lesion according to the moment when patients started the Fabry-specific therapy (before and after kidney biopsy). Our experience suggests that a revision of the national protocol is necessary by using early adding new indicators of kidney involvement in order to promote earlier specific treatment in FD females. In our opinion, histological evaluation through kidney biopsy, if available, should be included among the renal criteria for FD-specific therapy initiation.

According to the European expert consensus statement on therapeutic goals in FD, therapeutic renal objectives should be individualized based on the initial degree of albuminuria and severity of kidney failure [42]. The major challenge for physicians is to assess the prognosis of the disease as accurately as possible and to intervene therapeutically as early as possible. Kidney biopsy is the only renal biomarker that brings comprehensive information about kidney damage in Fabry disease. Thus, kidney biopsy and histological assessment can be used as the gold standard for evaluation of kidney involvement in FD and as an indicator of treatment initiation. 

In addition, due to the multiple organ systems affected by FD and the complexity of disease management, a multidisciplinary team approach is recommended; this should include nephrologists, cardiologists, neurologists and ophthalmologists in order to comprehensively evaluate the patients for the early diagnosis of organ damage, enabling the appropriate treatment to be decided and all therapeutic goals established. 

The current study limitations are the small sample size, its retrospective observational design and it being a single-center study.

## 5. Conclusions

Our findings showed that glomerular segmental sclerosis is the histological feature that predicted the long-term outcomes of progression towards kidney disease deterioration and mortality, and may be used as an early indicator of long-term outcome in FD patients.

The data from our small cohort underline the importance of kidney biopsy for the detection of early kidney involvement and provide additional support to the early initiation of FD-specific therapy, potentially improving the long-term outcome. Thus, proof of specific FD renal lesions as revealed by kidney biopsy could become a distinct criterion to initiate FD therapy, in the absence of other criteria according to current guidelines and different national protocols. Future studies are necessary in order to specify the role of renal histology in the establishment of the proper timing to start the FD treatment. 

Although clearly determining the optimal moment to initiate FD-specific treatment can be a real challenge in the case of certain patients (young, classical and non-classical female and non-classical male patients), there are definitive and essential tools that can help physicians to assess decide the best time to do that. The experience of our center showed that histologic evaluation of kidney involvement is one of these tools.

## Figures and Tables

**Figure 1 biomedicines-10-01520-f001:**
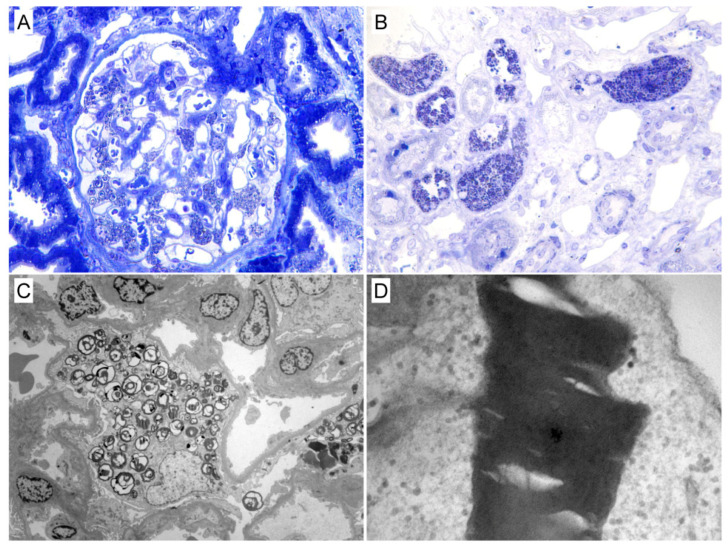
Kidney biopsy findings from patient No. 11. (**A**), Light microscopy image of a semi-thin plastic section showing a glomerulus with many lamellar inclusion bodies in podocytes (toluidine blue stain, 200×); (**B**), Light microscopy image of a semi-thin plastic section showing renal medulla with tubular epithelial cells and vascular smooth muscle cells having their cytoplasm filled with inclusion bodies (toluidine blue stain, 200×); (**C**), Electron microscopy image showing electron dense, lamellate inclusions (typical “zebra bodies”) in the cytoplasm of podocytes (5700×). (**D**), Electron microscopy image showing the densely packed lamellate membranes contained into a lysosome (24,000×).

**Figure 2 biomedicines-10-01520-f002:**
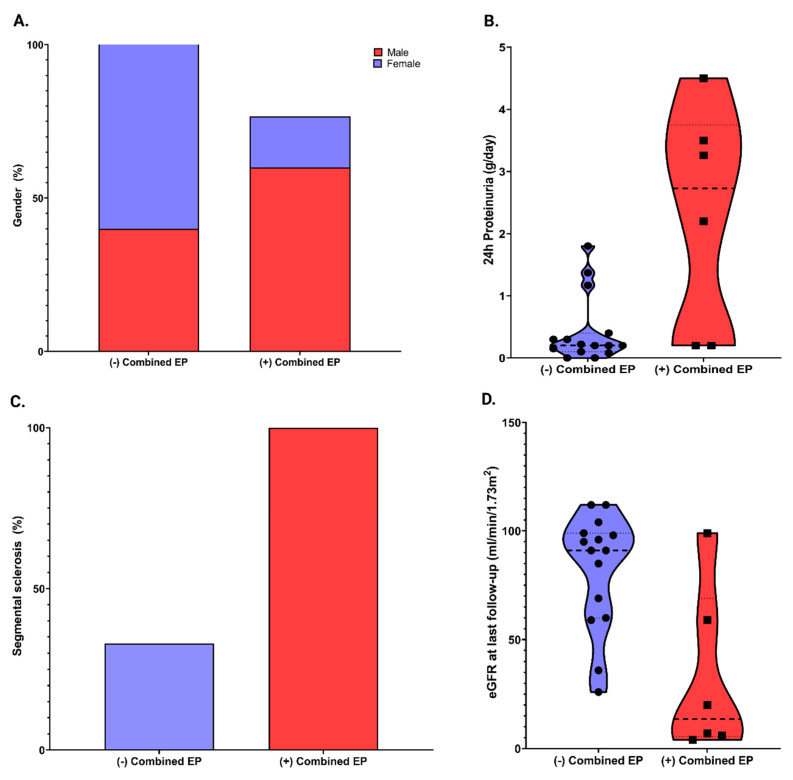
Variables associated with combined endpoint (EP). (**A**) Gender (*p* = 0.06); (**B**) Baseline 24 h proteinuria (*p* = 0.02); (**C**) Segmental sclerosis (*p* = 0.009); (**D**) eGFR at last follow-up (*p* = 0.003).

**Figure 3 biomedicines-10-01520-f003:**
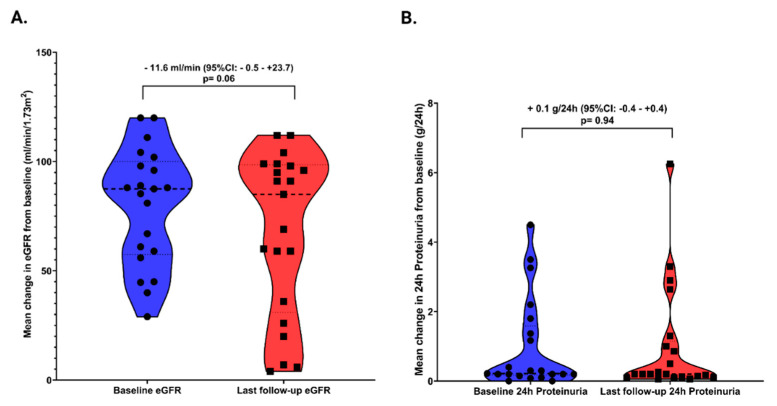
(**A**) Mean change in eGFR; (**B**) 24 h Proteinuria from baseline.

**Figure 4 biomedicines-10-01520-f004:**
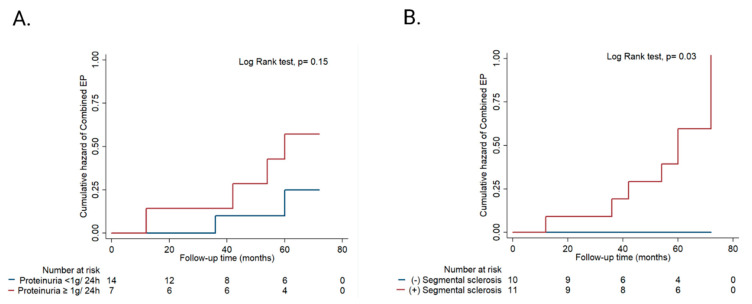
Cumulative hazard of the combined endpoint (EP) according to baseline: (**A**) Proteinuria ≥ 1 g/24 h; (**B**) segmental sclerosis.

**Table 1 biomedicines-10-01520-t001:** Baseline characteristics of the study cohort.

Patient No.	Sex/Age (yr)	HTN/DM	Mutation	eGFR (mL/min/1.73 m^2^)	CKD Stage	UACR (mg/g)	Proteinuria (g/24 h)	RAAS Blocker	ERT Duration (mo)	Cardiac Involvement	Neurologic Manifestation
1	M/17	+/−	c.779G > A	44.7	3	500	3.26	+	0	−	+
2	M/10	−/+	c.797A > C	87.5	2	10	0.08	−	0	+	−
3	M/41	−/−	c.863C > A	45	3	100	1.37	+	0	+	+
4	M/26	−/−	c.836A > G	89	2	345	0.3	−	0	−	+
5	M/44	−/−	c.644A > G	29	4	600	3.5	−	0	−	−
6	M/39	−/−	c.671delA	67	2	80	0.15	+	0	+	+
7	M/29	−/−	c.797A > C	120	1	10	0.2	−	0	−	+
8	F/50	+/−	c.797A > C	56	3	100	0.4	+	0	+	+
9	F/55	+/−	c.1224del66	61	2	819	1.8	+	0	+	+
10	F/49	−/−	c.779G > A	104.2	1	100	0.2	−	0	−	+
11	F/46	−/−	c.797A > C	96	1	10	0.1	−	0	−	+
12	F/30	−/−	c.797A > C	88	2	10	0.2	−	0	−	−
13	F/35	−/−	c.295C > T	81	2	20	0.3	−	0	−	−
14	F/63	+/−	c.1228A > G	59	3	30	0.22	+	0	+	+
15	F/35	+/−	c.485G > A	40	3	900	4.5	−	0	−	+
16	F/61	+/+	c.141G > A	85.3	2	20	ND	+	0	+	+
17	M/43	−/−	c.836A > G	111	1	300	1.17	+	12	−	+
18	M/32	−/−	c.1121_1123delAAG	120	1	20	ND	−	144	+	+
19	M/58	−/−	c.295C > T	102	1	10	0.2	−	72	−	+
20	M/37	+/−	c.485G > A	98	1	500	2.2	+	120	+	+
21	F/57	+/−	c.485G > A	88	2	150	0.2	+	27	−	+

HTN = arterial hypertension; DM = diabetes mellitus; eGFR = estimated glomerular filtration rate; UACR = urine albumin/creatinine ratio; ND = not detectable; RAAS = renine-angiotensin-aldosteron system; ERT= enzyme replacement therapy.

**Table 2 biomedicines-10-01520-t002:** Differences between group 1 (negative for the combined endpoint) and group 2 (reaching the combined endpoint) of patients.

Variables		Overall (*n* = 21)	(−) Combined Endpoint (*n* = 15)	(+) Combined Endpoint (*n* = 6)	*p* Value
**General** **features**	Age at baseline, years, mean ± SD	43.7 ± 14.2	45.7 ± 14.9	36.7 ± 9.5	0.27
	Age at diagnosis, years, mean ± SD	36.1 ± 14.6	39.5 ± 15.1	27.7 ±10.1	0.10
	Gender, *n* (%)				0.06
	Male	11 (52.4%)	6 (40%)	5 (83.3%)	
	Female	10(47.6%)	9 (60%)	1 (16.7%)	
	Hypertension, *n* (%)	8 (31.8%)	5 (33.3%)	3 (50%)	0.63
	Diabetes, *n* (%)	2 (9.5%)	2 (13.3%)	0 (0%)	0.9
	Obesity *n* (%)	3 (14.3%)	2 (13.3%)	1 (16.7%)	1
	Dyslipidemia, *n* (%)	11 (52.4%)	8 (53.3%)	3 (50%)	1
	Stroke, *n* (%)	4 (19%)	4 (26.7%)	0 (0%)	0.28
	Heart failure, *n* (%)	9 (42.9%)	7 (46.7%)	2 (33.3%)	0.44
	eGFR at baseline, ml/min/1.73 m^2^, mean ± SD	79.6 ± 26.7	82.5 ± 21.3	72.2 ± 38.7	0.44
	eGFR at last follow-up, ml/min/1.73 m^2^, mean ± SD	68.0 ± 37.3	82.2 ± 26.4	32.5 ± 38.5	0.003
	UACR at baseline, mg/g, median (IQR)	100 (15–422.6)	80 (20–300)	500 (10–675)	0.26
	24 h proteinuria at baseline, g/24 h, median (IQR)	0.2 (0.1–1.5)	0.4 (0.1–0.4)	2.7 (0.2–3.7)	0.02
	Follow-up period, months, mean ± SD	47.7 ± 19.1	49.2 ± 19.7	44.0 ± 18.4	0.58
**Fabry** **features**	Total MSSI at baseline, points, median (IQR)	22 (10.5–25)	22 (10–25)	22 (12.7–24.5)	0.91
	Neurologic manifestation at baseline, *n* (%)	18 (85.7%)	13 (86.6%)	5 (83.3%)	0.5
	Hypertrophic cardiomyopathy at baseline, *n* (%)	9 (42.8%)	7 (46.6%)	2 (33.3%)	0.47
	Arrhythmias at baseline (%)	8 (38.1%)	7 (46.7%)	1 (16.7%)	0.33
	Pacemaker at baseline (%)	1 (4.8%)	1(6.7%)	0 (0%)	1
	αGAL A α-GLA activity, nmol/h/mg, median (IQR)	0.4 (0.1–1.2)	0.6 (0.2–1.5)	0.3 (0–0.4)	0.13
	Plasma Llyso-GL3 at baseline, ng/mL, median (IQR)	7.6 (5.5–34.6)	6.8 (4.9–22.5)	27 (9.2–105.5)	0.22
**Kidney** **biopsy**	Glomeruli number, median (IQR)	8 (5–10)	8 (5–12)	6 (4–8)	0.15
	Segmental sclerosis, *n* (%)	11 (52.4%)	5 (33%)	6 (100%)	0.009
	Global sclerosis, *n* (%)	6 (28.6%)	3 (20%)	3 (50%)	0.29
	Interstitial fibrosis, *n* (%) <25% 25–50%	11 (52.4%) 10 (47.6%)	9 (60%) 6 (40%)	2 (33.3%) 4 (66.7%)	0.36
	Tubular atrophy, *n* (%)	9 (42.9%)	5 (33.3%)	4 (66.7%)	0.33
	Arteriopathy, *n* (%)	8 (38.1%)	5 (33%)	3 (50%)	0.63
	Podocyte GL3 deposits, *n* (%)	21 (100%)	15 (100%)	6 (100%)	1
	Tubular GL3 deposits, *n* (%)	21 (100%)	21 (100%)	21 (100%)	1
	Glomerular endothelial cell GL3 deposits, *n* (%)	20 (95.2%)	14 (93.3%)	6 (100%)	1
**Treatment**	FD-specific therapy at the moment of KB, *n* (%)	5 (23.8%)	3 (20%)	2 (33%)	0.6
	FD-specific therapy after KB, *n* (%)	20 (95.2%)	14 (93.3%)	6 (100%)	1
	FD-specific therapy duration, months, median (IQR)	36 (13–52.5)	37 (12.2–49.5)	30 (12.7–60)	0.98
	RAAS inhibitors, *n* (%)	10 (47.6%)	7 (46.7%)	3 (50%)	1
**Outcomes**	50% decrease in eGFR, *n* (%)	5 (23.8%)	0 (0%)	5 (83.3%)	<0.001
	ESKD KF, *n* (%)	3 (14.3%)	0 (%)	3 (50%)	0.01
	Mortality, *n* (%)	2 (9.5%)	0 (0%)	2 (33.3%)	0.07
	Combined endpoint, *n* (%)	6 (28.6)	−	−	−

(−) = absent; (+) = present; *n* = number; SD = standard deviation; eGFR = estimated glomerular filtration rate; UACR = urine albumin: creatinine ratio; IQR = interquartile range; % = percentage; MSSI = Mainz Severity Score Index; α-GLA = α-galactosidase A; lyso-GL3 = globotriaosylsphingosine; GL3 = globotriaosylceramide; FD = Fabry disease; KB = kidney biopsy; KF = kidney failure.

**Table 3 biomedicines-10-01520-t003:** Genotype, baseline level of α-GLA activity and lyso-GL3, as well as MISSI score of the study cohort.

Patient No.	Sex/Age (yr)	Mutation	Variant According to ACMG	Type of Mutation	Phenotype	α-GLA Activity (Reference Range > 2.8 µmol/L/h)	Lyso-GL3 (Reference Range 0–3.5 ng/mL)	MSSI Score
1	M/17	c.779G > A	Pathogenic	Missense	Classic	0.5	NA	26
2	M/10	c.797A > C	Pathogenic	Missense	Classic	0	101.1	9
3	M/41	c.863C > A	Pathogenic	Missense	Classic	0.3	98	24
4	M/26	c.836A > G	Pathogenic	Missense	Classic	0.13	22.5	25
5	M/44	c.644A > G	Pathogenic	Missense	Late-onset	0.4	34.6	20
6	M/39	c.671delA	Pathogenic	Deletion	Classic	2	5.5	32
7	M/29	c.797A > C	Pathogenic	Missense	Classic	0	129.2	9
8	F/50	c.797A > C	Pathogenic	Missense	Classic	1.2	6.7	33
9	F/55	c.1224del66	Pathogenic	Deletion	Classic	1	7.6	22
10	F/49	c.779G > A	Pathogenic	Missense	Classic	NA	6.8	3
11	F/46	c.797A > C	Pathogenic	Missense	Classic	0.4	10.7	14
12	F/30	c.797A > C	Pathogenic	Missense	Classic	0.7	3.9	10
13	F/35	c.295C > T	Pathogenic	Nonsense	Classic	1.5	4.4	7
14	F/63	c.1228A > G	Pathogenic	Missense	Classic	1.8	4.9	42
15	F/35	c.485G > A	Pathogenic	Nonsense	Classic	0.32	5.8	14
16	F/61	c.141G > A	Pathogenic	Nonsense	Classic	3.3	3.3	13
17	M/43	c.836A > G	Pathogenic	Missense	Classic	0.1	42.3	25
18	M/32	c.1121_1123delAAG	Pathogenic	Deletion	Classic	0.5	16.8	22
19	M/58	c.295C > T	Pathogenic	Nonsense	Classic	0.1	NA	24
20	M/37	c.485G > A	Pathogenic	Nonsense	Classic	0.1	19.4	24
21	F/57	c.485G > A	Pathogenic	Nonsense	Classic	0.3	6.3	11

α-GLA = α-galactosidase A; lyso-GL3 = globotriaosylsphingosine; MSSI = Mainz Severity Score Index; NA = not available; ACMG = American College of Medical Genetics and Genomics.

**Table 4 biomedicines-10-01520-t004:** Light microscopy and electron microscopy findings of the renal biopsy specimens.

Patient No.	Sex/Age (yr)	Global Sclerosis	Segmental Sclerosis	Glomerular Hyaline	Interstitial Fibrosis	Tubular Atrophy	Arterio- Pathy	Podocyte GL3 Deposits	Tubular GL3 Deposits	Glomerular Endothelial Cell GL3 Deposits
1	M/17	+	+	−	+	NA	+	+	+	+
2	M/10	−	−	−	−	−	−	+	+	+
3	M/41	−	−	−	+	NA	+	+	+	+
4	M/26	−	−	−	−	−	−	+	+	+
5	M/44	−	+	+	+	+	+	+	+	+
6	M/39	+	+	+	+	+	−	+	+	+
7	M/29	−	+	−	−	−	−	+	+	+
8	F/50	+	+	−	+	+	+	+	+	+
9	F/55	+	+	+	+	+	+	+	+	+
10	F/49	−	−	−	−	+	−	+	+	+
11	F/46	−	−	−	−	−	−	+	+	+
12	F/30	−	−	−	−	−	−	+	+	+
13	F/35	−	−	−	−	−	−	+	+	+
14	F/63	−	−	−	−	NA	−	+	+	−
15	F/35	+	+	−	−	−	+	+	+	+
16	F/61	−	−	−	+	NA	+	+	+	+
17 *	M/43	−	+	−	+	+	+	+	+	+
18 *	M/32	−	−	−	−	−	−	+	+	+
19 *	M/58	−	+	−	+	+	−	+	+	+
20 *	M/37	+	+	−	+	+	−	+	+	+
21 *	F/57	−	+	+	−	−	−	+	+	+

* Previously enzyme replacement therapy-treated patients; Plus (+) sign represents the presence and minus (−) sign represent the absence; M = male, F = female; NA = not applicable; GL3 = globotrioasylceramide.

**Table 5 biomedicines-10-01520-t005:** Subgroup comparison analysis between patients who started the Fabry specific therapy before and after the kidney biopsy.

Variables	Treatment before KB (*n* = 5)	Treatment after (KB) (*n* = 16)	*p* Value
Combined EP, *n* (%)	2 (40%)	4 (25%)	0.58
Segmental sclerosis, *n* (%)	4 (80%)	7 (44%)	0.30
Global sclerosis, *n* (%)	1 (20%)	5 (31.3%)	1
Interstitial fibrosis, *n* (%) <25% 25–50%	2 (40%) 3 (60%)	9 (56.3%) 7 (43.8%)	0.63
Tubular atrophy, *n* (%)	3 (60%)	6 (37.5%)	0.61
Arteriopathy, *n* (%)	1 (20%)	7 (43.8%)	0.60
Podocyte GL3 deposits, *n* (%)	5 (100%)	16 (100%)	1
Tubular GL3 deposits, *n* (%)	5 (100%)	16 (100%)	1
Glomerular endothelial cell GL3 deposits, *n* (%)	5 (100%)	16 (100%)	1

*n* = number; GL3 = globotriaosylceramide; KB = kidney biopsy; % = percentage.

## Data Availability

The data presented in this study are available on request from the corresponding author.

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
