# Peer review of "The Impact of Kidney Biopsy for Fabry Nephropathy Evaluation on Patients’ Management and Long-Term Outcomes: Experience of a Single Center"

_biomedicines, 2022, doi:10.3390/biomedicines10071520_

Round 1

Reviewer 1 Report

Thank you to the authors for submitting this manuscript describing 21 patients with Fabry Disease and an analysis of their clinical features and kidney histopathology compared to clinical outcomes. I have some queries:

1) Please confirm to the KDIGO consensus nomenclature for kidney disease, ie, "Kidney Failure" rather than ESKD, etc

2) Please review the length and structure of some sentences throughout as these tend to be long and confusing at times. An example in the introduction is Lines 64-67. This detracts from your key messages.

3) Please list the ACMG criteria for each variant that substantiate that they are ACMG Class 5 (Pathogenic). 

4) There is some confusion in regards to how the cohort is divided according to the prespecified combined endpoint. When is that combined endpoint being assessed? Is it at most recent follow up? Is it at a standardised time point (2yrs/5yrs/etc) post kidney biopsy? Please clarify this substantially. This is a major point to revise. 

5) The baseline of the study must be the time of first kidney biopsy given that the analysis is largely geared from that. It is important to stratify out those patients who had already been exposed to FD treatment (ie, ERT) prior to that kidney biopsy. As such, please analyse those patients with FD treatment prior to kidney biopsy separately to those who were ERT naive at baseline/biopsy. This is a major point to revise. 

6) There are some sound key messages here that need to be stated much more clearly, specifically that greater proteinuria at baseline and/or greater segmental sclerosis at baseline are predictors of worse kidney outcomes. There is lots of good content in the manuscript but at times it becomes so voluminous as to almost obscure your key findings and messages. Please review/edit to reclarify and uplift the key messages. 

Reviewer 2 Report

In this paper, authors investigated the influence of some pathological findings on the management of Fabry disease patients, looking for the association between general and FD specific features and long-term patients' outcome. The paper is interesting and the topic on a so rare disease as FD could be of great interest in this contest. Authors conclude that histological evaluation is an important tool for the detection of early kidney involvement and provides additional support to early initiation of FD specific therapy. Some comments:

- it is known that FD involved also the heart, however authors did not mention this aspect. Could we have some more data oh this? and iof not available, could they add this limitation?

- it is important to stress that FD patients have to be referred to a series of specialistis after tha diagnosis, such as cardiologists, oculistics, neurologists...please remind that in the Discussion;

- considering the importance of an early diagnosis to start as soon as possibile ERT, it could be interesting to refere to "Di Nora C. Heart transplantation in cardiac storage diseases: data on Fabry disease and cardiac amyloidosis. Curr Opin Organ Transplant. 2020 Jun;25(3):211-217. doi: 10.1097/MOT.0000000000000756", where an alternative solution has been proposed in case of severe heart involvement;

- I would suggest to better define the aims of the paper according to the results presented 

Reviewer 3 Report

1. Please specify treatment for fabry disease. there are now newer agent Galafold® (migalastat). So, please specify them all.

2.  How many subgroup patients received kidney transplantation? How many were males vs females? Did they have recurrence or graft failure after transplantation?

3. Any data on follow up biopsy after treatment

4. How many were on ACEIs/ARBs? 

Round 2

Reviewer 1 Report

Thank you to the authors for this revised manuscript. The manuscript is approved but there remain several issues that must be addressed in the view of this reviewer:

1) There remain English language syntax and grammatical errors throughout.

2) The specific and individual ACMG criteria for each individual variant's ACMG variant classification must be listed. Further, it must be listed as to which participant/s each variant is attributed.

3) I reaffirm strongly that the baseline of the study must be the time of first kidney biopsy. Further, those patients who have been exposed to FD treatment prior to that biopsy must be analysed separately as the effects of that FD treatment on histology cannot be otherwise be controlled for. It is a major methodological concern to not fully address this. 

Reviewer 2 Report

The authors have clearly answered to all my previous doubts. The paper is improved in this current form, I would like to accept for publication. 

Author Response

Thank you very much!

Round 3

Reviewer 1 Report

I have no further queries.